# Advances in the Production of Theanine by Plants and Microorganisms

**Shujian Xiao** [1,†], **Rong Qian** [1,†], **Shunyang Hu** [1], **Zhongdan Fu** [1], **Ting Bai** [1], **Wei Wang** [1], **Jie Cheng** [1,*]
and **Jiamin Zhang** [2,*]

1   College of Food and Bioengineering, Chengdu University, Chengdu 610106, China;
    xiaoshujian1995@163.com (S.X.); qianrong@stu.cdu.edu.cn (R.Q.); hushunyang@stu.cdu.edu.cn (S.H.);
    fzd1666@163.com (Z.F.); baiting@cdu.edu.cn (T.B.); wangwei8619@163.com (W.W.)
2   Meat Processing Key Laboratory of Sichuan Province, Chengdu 610106, China
*   Correspondence: chengjie@cdu.edu.cn (J.C.); zhangjiamin@cdu.edu.cn (J.Z.)
†   These authors contributed equally to this work.

**Abstract:** Theanine, a representative non-protein amino acid in tea, is one of the umami components of tea and a major factor in the formation of the unique flavor of tea leaves. In addition to its delicious taste, theanine has a variety of health functions and is used in the food supplement, pharmaceutical, nutraceutical, and cosmetic industries. This review briefly describes the physiological functions, application prospects, and production methods of theanine. The biosynthetic pathway of L-theanine in natural plants is also introduced. Finally, the microbial synthesis of L-theanine is divided into two major biosynthetic pathways guided by glutamine and glutamate according to the different substrates. According to the status, at present, of the microbial synthesis of L-theanine, the future development of microbial synthesis of L-theanine is prospected, in order to provide technical and theoretical basis for in-depth research on the industrial production of theanine.

**Keywords:** theanine; physiological function; food supplement; microbial synthesis

## 1. Introduction

Theanine is one of the umami components of tea and a very representative non-protein amino acid in tea tree. L-theanine was first isolated and extracted from tea leaves and its chemical molecular structure formula was determined in 1950 [1]. Theanine possesses numerous physiological benefits and is widely used as an additive in the fields of food, dietary supplements, pharmaceuticals, health products, and cosmetic sectors (Figure 1).

In medicinal health, theanine has a wide range of biological activities, mainly in its synergistic ability to protect nerves, relieve stress, lower blood pressure, improve immunity, anti-aging, anti-inflammatory and anti-tumor. Williams et al. reported that L-theanine as an oral nutrient at 200–400 mg can have a significant effect on reducing stress and anxiety [2]. L-theanine may prevent bladder dysfunction by inhibiting chronic sympathetic hyperactivity [3]. As a unique bioactive regulator, theanine can inhibit tumor growth and metastasis by mediating cell apoptosis [4,5]. In the cosmetics industry, theanine has anti-aging and moisturizing functions. In recent years, in vitro functional and safety evaluation studies have been conducted on a series of products to clarify the efficacy and mechanism of theanine as an active ingredient in cleansing, antibacterial, and skincare products. A wide range of skincare and make-up products containing theanine are on the market, at present [6]. In addition, because of the numerous benefits that theanine provides to human health, it has been researched and applied as a food additive in foods for a long time [7]. Since 1964, theanine has been considered as one of the common functional food additives in Japan, allowing the addition of theanine to foods other than infant foods. Based on the good safety and stability of theanine, it can be used as a food flavor regulator to reduce

bitterness and improve the freshness of tea soup [8]. Theanine is most widely used in food products, such as tea drinks, baked snacks, and frozen sweets [9].

**APPLICATION OF THEANINE**

**Figure 1.** Application of L-theanine in different fields.

Due to the application value of theanine in many fields, the estimated market value of theanine reached USD 50 million as early as 2020 [10]. With the development of downstream industries and the maturation of theanine application technology, the global theanine market demand has continued to increase in recent years [11]. However, there are very few natural sources of theanine, and the direct extraction of theanine from tea leaves has relatively high running costs, many extraction processes, low extraction rates, and contains other impurities that are difficult to purify [12]. Early research focused on the synthesis or isolation and purification of theanine by chemical methods; however, this method has the disadvantages of cumbersome steps, difficult separation, high cost, and unsatisfactory economy, and the products often contained D-theanine [13]. The pharmacokinetics and metabolic behavior of D-theanine are different from L-theanine, and the presence of D-theanine can inhibit the absorption of theanine, leading to the metabolism of L-theanine to produce ethylamine. Due to the toxicity of ethylamine and the use of some hazardous chemicals in chemical synthesis, people have a skeptical attitude towards the safety of the chemical synthesis of theanine [7,14]. In contrast, microbial synthesis is the most promising production method at this stage, which ensures the safety and purity of the target product while significantly increasing economic efficiency [12,15]. In addition, the microbial production of L-theanine can replace traditional non-environmentally friendly and energy-intensive methods, and is worth trying to apply in various industries [10,16].

In this review, we focus on the present-day discovery of the biosynthetic pathway of L-theanine in natural plants. We analyze and summarize the microbial synthesis methods, research needs, and future development trends of L-theanine, in order to provide theoretical support for the industrialization of theanine.

## 2. The Biosynthetic Pathway of L-Theanine in Plants

### 2.1. The Biosynthetic Pathway of L-Theanine in Tea Tree

Theanine is one of the secondary metabolites that are significantly represented in tea tree. Theanine is present in all organs of the tea tree, except the fruit, the highest content of which is in the young leaves, followed by the roots, old leaves, stem bark, and stem xylem in adult tea trees [6]. The precursors for the synthesis of theanine were determined to be ethylamine and glutamate by isotopic tracing with $^{14}$C in 1974 [17]. Recently, Cheng et al. [18], again using isotopic labeling, confirmed that the presence of ethylamine is the reason for the different levels of theanine accumulation in various plants. In addition, research have shown that ethylamine is specifically synthesized by the root tissue of tea trees, utilizing the genes of the tea tree itself [19].

Research key enzymes and genes for L-theanine synthesis and metabolism in tea tree are beneficial for achieving the large-scale microbial synthesis of L-theanine. Alanine decarboxylase (AlaDC) has been identified, which can catalyze the decarboxylation of L-alanine to produce ethylamine in vitro [20]. However, there is no direct evidence that ethylamine in tea tree is produced by the decarboxylation of alanine. Early researchers speculated through experiments that alanine is a precursor of ethylamine [17]. However, a recent experiment has refuted this claim [19]. According to isotope labeling, it has been found that L-alanine is a precursor of L-glutamate and L-glutamate is produced from L-alanine through transamination, which in turn participates in the synthesis of L-theanine [19]. The formation of ethylamine in tea tree is still unclear [7]. Initially, Bai et al. [21] cloned the gene for serine decarboxylase (SDC), one of the important enzymes in the theanine synthesis pathway, from tea tree that also has the ability to produce alanine decarboxylation. However, since decarboxylation is highly restricted in bacteria, the use of AlaDC in the bacterial engineering of the theanine production pathway is not feasible [22]. In addition to AlaDC and SDC, enzymes related to the theanine synthesis pathway containing theanine synthase (TS), glutamine synthetase (GS), arginine decarboxylase (ADC), glutamate dehydrogenase (GDH), and glutamate synthase (GOGAT) [20,23,24]

TS and theanine hydrolase are key enzymes in theanine biosynthesis and metabolism, which are specific for theanine metabolism [25]. The theanine was produced from ethylamine and L-glutamate by TS and GS in the presence of ATP and metal ions ($Mg^{2+}$ and $K^+$) [26]. Wei et al. [27] found five GS gene sequences from the tea tree genome. GS is divided into two isoenzymes, GS1 and GS2, located in the chloroplast and cytoplasm. GS1 is mainly involved in the synthesis of nitrogen in the roots, while GS2 mainly synthesizes $NH^{4+}$ of glutamine by photorespiration [28]. The aminamide-$\alpha$-ketoglutarate aminotransferase (GOGAT, glutamate synthase), which converts glutamine and $\alpha$-ketoglutarate into two molecules of glutamate. One molecule of glutamate can be used as a substrate for GS and the other molecule of glutamate can be used for the synthesis of nitrogenous compounds [23]. Glutamate dehydrogenase (GDH) can assimilate $NH_3$ and $\alpha$-ketoglutarate into glutamate, which requires NADH [7,29]. ADC plays a key physiological role in steps, such as budding and boll differentiation [30]. The theanine synthesis pathway in tea tree is shown in Figure 2.

### 2.2. The Biosynthetic Pathway of L-Theanine in Other Plants

Apart from the *Camellia. sinensis*, which is able to synthesis theanine entirely from its own metabolism, other plants can only synthesis theanine almost exclusively from exogenous substrates [16]. The metabolites in *C. sinensis*, *C. nitidissima, C. japonica, Z. mays, Arabidopsis thaliana*, and *Solanum lycopersicum* were detected. The results show that glutamate accumulates in all of the abovementioned plants; however, only *Camellia* plants and *Z. mays* can accumulate ethylamine and theanine, with the greatest accumulation of theanine in *C. sinensis* [18]. When isotope-labeled ethylamine was supplied to the roots and leaves of the six plants mentioned above, theanine was detected in all of these plants; this leads to the inference that enzymes/genes capable of catalyzing theanine synthesis are present in all these plants [18,19]. In addition, since GS is not a specific enzyme

exclusively belonging to tea tree, some plants can also synthesize L-theanine from GS when the substrate is suitable [31,32]. However, due to the fact that the biosynthesis of L-theanine mainly comes from tea tree, most research has focused on tea tree, and there is relatively little research on the synthesis of theanine by other plants [7]. The limitation at present lies in the lack of characterization of the biosynthesis of L-theanine in tea tree, which is also based on the limitation of the amount of theanine synthesis in tea tree [33]. It is precisely because the market demand, at present, for theanine exceeds the supply that the research on the metabolism of L-theanine continues to expand, and the large-scale production of L-theanine has become one of the main hotspots for research [34].

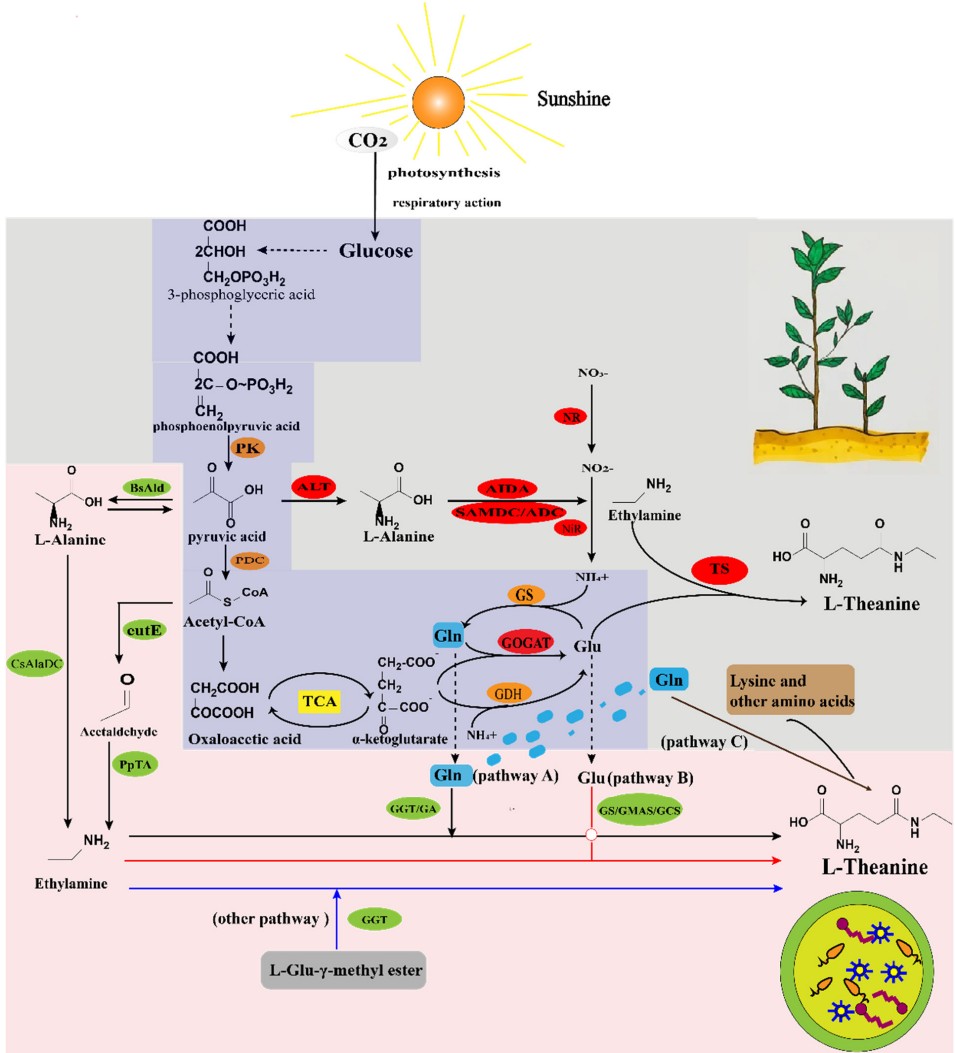

**Figure 2.** Synthetic pathways of theanine in tea plants and microorganisms. The light−gray background image represents the L−theanine production pathway in tea tree; the light−pink background image represents the microbial L−theanine production pathway; the purple section shows the co−synthetic pathway. Glu: glutamate; Gln: glutamine; main enzymes involved in the metabolic pathway; PK: pyruvate kinase; PDC: pyruvate decarboxylase; ALT: alanine amino-transferase; AIDA: alanine decarboxylase. SAMDC: S−methionine decarboxylase; ADC: argi-nine decarboxylase; NR: nitrate reductase; NiR: nitrite reductase; TS: theanine synthase; GOGAT: glutaminase−α−ketoglutarate aminotransferase; GDH: glutamate dehydrogenase; BsAld: alanine dehydrogenase; CsAlaDC: alanine decarboxylase; eutE: acetaldehyde dehydrogenase; pPTA: amino-transferase; GGT: gamma−glutamyl transpeptidase; GA: glutaminase; GMAS: gamma−glutamine methyl amide synthase; GS: glutamine synthetase; GCS: gamma−glutamyl cysteine synthase; L−Glu−γ−methyl ester: L−glutamyl−γ−methyl ester.

## 3. The Synthesis Pathway of L-Theanine in Microorganisms

To date, there are mainly two biosynthetic routes of L-theanine in microorganisms (Figure 2). One is the glutamate-mediated pathway, which uses glutamate (Glu) as a precursor [35] and requires ATP for energy supply, as shown in pathway A in Figure 2. The other is the glutamine-mediated pathway, which uses glutamine (Gln) as a precursor and is based on the generation of γ-glutamyl transfer reactions [36] (refer to pathway B and pathway C in Figure 2). To achieve the microbial synthesis of L-theanine, it is theoretically possible to use TS of the tea tree. However, TS is an enzyme that is easily inactivated and very unstable in vitro in tea tree, is ATP-dependent, and is difficult to isolate and purify as well as to characterize [37]. This characteristic of it dictates the need to study other microbial enzymes applicable to the microbial synthesis of L-theanine. Several microbial enzymes with L-theanine production capacity were identified. They are glutamine synthetase (GS), gamma-glutamine methylamide synthetase (GMAS), gamma-glutamylcysteine synthetase (γ-GCS), gamma-glutamyl transpeptidase (GGT), and L-glutaminase [7,33]. Representative examples of glutamate and glutamine-mediated products of L-theanine are detailed in Table 1, which includes types of microbial processing routes, types of strains, and characteristics of production strategies, typical biomass species, production potency, and yields.

**Table 1.** Types of microbial processing routes, strain species and characteristics, typical biomass species, production potential, yield, and references for L-theanine synthesis mediated by glutamate, glutamine, and other substrates.

| | Host | Engineered Strategy | L-Theanine Titer (g/L) | L-Theanine Yield (g/g) | Substrate | Reference |
|---|---|---|---|---|---|---|
| Pathway (A) | *Micrococcus glutamicus* | Coupling of cell-free extracts of baker's yeast with partially purified glutamine synthetase | 0.01 | – | Sodium glutamate | [38] |
| Pathway (A) | *E. coli* | Expression of GS gene | 29.6 | 0.85 | Sodium glutamate | [28] |
| Pathway (A) | *E. coli* | Expression of GS gene | 15.3 | 0.45 | Sodium glutamate | [26] |
| Pathway (A) | *E. coli* | Whole-cell catalytic reaction | 6.2 | 0.178 | Sodium glutamate | [39] |
| Pathway (A) | *E. coli* | Targeted transformation modification of γ-GCS mutants | 30.4 | 0.871 | Sodium glutamate | [40] |
| Pathway (A) | *E. coli* | Recombinant GMAS coupled to an ATP regeneration system | 110 | 1.00 | Sodium glutamate | [41] |
| Pathway (A) | *E. coli* | Co-expressions of PPK and GMAS | 34.67 | 0.66 | Sodium glutamate | [36] |
| Pathway (A) | *C. glutamicum* | Batch make-up fermentation strategy | 42.0 | 0.196 | Glucose | [42] |
| Pathway (A) | *E. coli* | Ethanolamine addition | 30.45 | 0.201 | Glucose | [43] |
| Pathway (A) | *E. coli* | Engineering of a one-step fermentation pathway from sugar and ethylamine | 70.6 | 0.42 | Glucose | [44] |
| Pathway (A) | *Pseudomonas putida* | Fermentation process using Pseudomonas aeruginosa metabolic engineering compatible with various alternative carbon sources | 17.2 | – | Glucose with xylose | [45] |
| Pathway (A) | *E. coli* | Establishing a TA pathway without ethylamine supplementation | 1.53 | – | Glucose | [46] |
| Pathway (B) | *P. nitroreducens* | Enzyme-catalyzed reaction | 47.034 | 0.458 | Glutamine | [47] |

**Table 1.** *Cont.*

| | Host | Engineered Strategy | L-Theanine Titer (g/L) | L-Theanine Yield (g/g) | Substrate | Reference |
|---|---|---|---|---|---|---|
| Pathway (B) | *P. nitroreducens* | Enzyme-catalyzed reaction | 85.358 | 0.661 | Glutamine | [48] |
| Pathway (B) | *Trichoderma koningii* | Enzyme-catalyzed reaction | 7.491 | 0.171 | Glutamine | [49] |
| Pathway (B) | *E. coli* | Enzyme-catalyzed reaction | 20.904 | 0.715 | Glutamine | [50] |
| Pathway (B) | *E. coli* | Crude enzyme catalysis | 26.884 | 0.578 | Glutamine | [51] |
| Pathway (B) | *E. coli* | Separation of pure enzyme catalysis | 12.124 | 0.964 | Glutamine | [52] |
| Pathway (B) | *E. coli* | Point mutation, pure enzyme catalysis | 18.622 | 0.89 | Glutamine | [53] |
| Pathway (B) | *E. coli* | Expression of GGT in recombinant *E. coli* using small ubiquitin-related modifier (SUMO) fusion technology | 41 | 0.800 | Glutamine | [54] |
| Pathway (B) | *B. subtlis* | Separation of pure enzyme catalysis | 32.66 | 0.74 | Glutamine | [55] |
| Pathway (B) | *C. glutamicum* | Separation of pure enzyme catalysis | 18.17 | 0.898 | Glutamine | [56] |
| Pathway (B) | *E. coli* | Whole-cell catalysis | 34.650 | 0.790 | Glutamine | [57] |
| Pathway (C) | *Luteibacter* | Separation of pure enzyme catalysis | $34.223 \times 10^{-6}$ | – | Glutamine | [58] |
| Other pathways | *E. coli* | Whole-cell catalysis | 15.3 | 0.95 | Glutamate-γ-methyl ester | [59] |

*3.1. Glutamate-Mediated Pathway of L-Theanine in Microorganisms*

3.1.1. Enzyme-Catalyzed Method

The synthases involved in the Glu-mediated pathway of L-theanine are GS, γ-GMAS, and γ-GCS, which catalyze the synthesis of L-theanine using Glu and ethylamine as substrates in the presence of ATP. Tachiki et al. found that GS in bacteria can be used as a biocatalyst for the production of L-glutamine [60]. Subsequently, the first study by the same group of researchers found that the GS of Micrococcus glutamicus ATCC 13032 could synthesize L-theanine when the substrate $NH^{4+}$ was replaced by ethylamine [38]. This study was the first to demonstrate the use of inexpensive L-glutamate and ethylamine, followed by the production of L-theanine by coupling baker's yeast preparations with GS in bacteria. The GS of *Pseudomonas aeruginosa* Y-30 was identified and successfully characterized the reactivity to theanine synthesis [61]; then, 29.6 g/L theanine was synthesized by optimizing the reaction system [62]. The isolated GS gene was then ligated into the expression vector pET21a and expressed in *E. coli* AD494(DE3). The enzyme productivity expressed in this system was 30-times higher than that in *P. taetrolens* Y-30 and its activity towards ethylamine was 7% higher than that of ammonia [28]. In addition, GS from other different bacterial sources were studied [26,39,63]; they were also consistently identified as having the ability to synthesize theanine, of which, Zhou et al. [26] used *Bacillus subtilis*-derived *GS* and 30 g/L yeast cells to catalyze the production of 15.3 g/L theanine from 200 mmol/L glutamate, with a conversion rate of 44%. Zhu et al. [39] constructed an engineered bacterium containing the GS gene of *Pseudomonas fluorescens*, whose enzyme activity was about 126.64 times that of the starting strain *E. coli* BL21 [39]. It catalyzed the reaction of sodium L-glutamate and ethylamine hydrochloride to produce 6.2 g/L theanine, and its ability to synthesize theanine was significantly improved compared to the starting strain *E. coli* BL21 [39]. Although the use of GS in bacteria produces a low level of L-theanine by-products, its low ethylamine reactivity is still unchangeable, which has led researchers to constantly search for catalase enzymes with high reactivity towards ethylamine.

The γ-GCS can make many other types of amino acids, such as alanine glutamylated [35]. Miyake et al. [35] first identified the ability of γ-GCS from *E. coli* to combine

glutamate and ethylamine for the production of theanine. The production of 2.1 g/L theanine was catalyzed by 414 mmol/L glutamate using γ-GCS and its own metabolized ATP; however, the yield was low at 2.9%. It is also worth exploring whether the theanine synthesized here can only be achieved in *E. coli* [35]. Some researchers have suggested that γ-GCS is expressed at a higher level compared to GS [40]. However, it must be noted that, in addition to the low yield, the formation of the by-product γ-glutamylalanine is also a disadvantage of this theanine production system. Therefore, altering the host glucose metabolic pathway or the substrate specificity of γ-GCS is necessary; however, the latter seems to be better achieved [35]. The crystal structure of γ-GCS in *E. coli* was determined and the Cys-binding site was identified [64]. Along this line of thought, Yao et al. [40] recently used a random mutagenesis approach for the targeted evolution of *E. coli* γ-GCS. Mutant γ-GCS13B6 increased the production of L-theanine and the catalytic efficiency of ethylamine by 14.6- and 17.0-fold, respectively, compared with the wild-type enzyme. It catalyzed 200 mmol/L glutamate and ethylamine to produce 30.4 g/L theanine with a conversion rate of 87%.

Kimura et al. [65] first purified approximately 70-fold GMAS from the methyl phage strain Methylo-phage sp. AA-30 in 1992. Studies have shown that the enzyme has maximum activity at pH 7.5 and 40 °C, which binds ethylamine to the γ-amino group of L-glutamate to synthesize L-theanine, demonstrating the broader substrate specificity of GMAS. To investigate this further, Yamamoto et al. [66] selected several strains from about 200 species of methylamine or methanol-assimilating bacteria for the research, and again demonstrated that the amount of theanine formed by GMAS was much greater than that formed by *E. coli* cells expressing the teatrolens Y-30 GS. The group then went on to investigate the production of L-theanine by GMAS of *Methylovorus mays* No. 9 through coupled fermentation based on yeast sugars. A total of 110 g/L L-theanine was synthesized in a 100% yield using recombinant GMAS as a catalyst in the optimized mixed system containing 40 g/L yeast cells, 600 mmol/L monosodium glutamate, 600 mmol/L ethylamine hydrochloride, and 30 U/mL GMAS enzyme solution [41].

### 3.1.2. Whole-Cell Catalytic Method

Although only inexpensive glutamate and an equivalent amount of ethylamine need to be added, ATP regeneration is still key in relation to the use of GMAS for the industrial production of L-theanine [34]. In addition, microbial conversion is a simpler and more convenient process than enzyme catalysis, which uses intact microbial cells to convert the substrate to L-theanine [29]. The establishment of an ATP regeneration system for PPK after the polyP-fed whole-cell biocatalytic synthesis of L-theanine using inexpensive monosodium glutamate and ethylamine hydrochloride as substrates and the optimum whole-cell catalytic reaction conditions were also determined, and the conversion of monosodium glutamate was 66.34% [36]. However, the examples described above all seem to involve only enzyme-catalyzed and microbial conversion methods for the synthesis of L-theanine, in which, although the cycle time is shorter [49], enzyme preparation and enzymatic reactions need to be conducted in steps, and the catalytic system is complex, cumbersome, and not efficiently recyclable [48]. In contrast, the microbial fermentation method has the advantages of the low cost of substrate raw material, one-step reaction, easily obtained product from the reaction solution, high conversion efficiency, and can be produced in large quantities, which makes it more suitable for scaling up to the industrial production of L-theanine [67].

### 3.1.3. Ethanolamine Flow plus Microbial Fermentation Method

Fan et al. [42] attempted to engineer the fermentative production of L-theanine in an industrially safe host, *Corynebacterium glutamicum*, and is the first example of supplementation with ethylamine to achieve the fermentative production of L-theanine. Compared with *C. glutamicum*, *E. coli* is used as a more easily designed host due to its good genetic background [68,69]. By introducing a heterologous GMAS from *Paracoccus aminovorans*

into *E. coli*, overexpressing natural citrate synthase, introducing glutamate dehydrogenase, pyruvate carboxylase, and phosphoenolpyruvate carboxykinase, as well as optimizing feeding and a series of other operations, the recombinant strain TH11 was able to produce 70.6 g/L of theanine, with a 42% conversion of glucose. This is also the first report on the engineering of a pathway for the production of theanine by fermentation in *E. coli* [44]. In addition, Benninghaus et al. [45] used *Pseudomonas putida* KT2440, a fast-growing strain with low nutrient requirements, as a starting strain for the production of theanine. Specifically, the use of a heterologous enzyme from *Methylorubrum extorquens* DM4 for L-theanine production and the overexpression of the *Caulobacter crescentus*-derived xylose manipulator xylXABCD using the pEV3 plasmid enabled the strain to utilize xylose as a carbon source. The recombinant strain Thea1 was able to produce 10 g/L of theanine from the recombination of glucose and xylose with a conversion rate of 3.3% to the carbon source when cultured in a batch replenishment bioreactor. It is also capable of producing 17.2 g/L of theanine from glycerol, with a conversion rate of 13% to glycerol. The recombinant strain TheaX was able to produce 21 g/L of theanine from the overlap of glycerol and xylose, with a conversion rate of 3.3% to the carbon source [45]. At present, this is the first L-theanine process to use *Pseudomonas aeruginosa* and the first to be compatible with the use of various alternative carbon sources [45].

### 3.1.4. One-Step Fermentation Method

Ethylamine is an essential starting material for the synthesis of L-theanine [46]; however, its use in the industrial production of L-theanine increases production costs and poses various drawbacks, such as a low boiling point (16.6 °C), toxicity, the need for special equipment for replenishment, which increases the complexity of production, and significant negative impact on human health and the natural environment [70]. In addition, the incomplete metabolism of ethylamine accumulates in high extracellular quantities, limiting the growth of the bacterium and weakening ATP regeneration, thereby inhibiting the synthesis of theanine. This has forced researchers to seek a way to build an efficient ethylamine synthesis pathway in cells to produce L-theanine [63,71]. In tea tree, ethylamine is mainly produced through the decarboxylation of alanine, and no synthetic pathway for ethylamine has been reported in microorganisms [72]. Therefore, researchers began to try to produce L-theanine without adding ethylamine.

Recently, Tabata and Shoto [73] designed an *E. coli* containing PP_5182, PP_0596, jm49_01725, and RFLU_RS03325 four enzyme genes. These enzymes exist in *Pseudomonas* and can use acetaldehyde and L-alanine as substrates to produce ethylamine. Finally, the recombinant *E. coli* was successfully fermented to produce 1.48 g/L of L-theanine. Then, a cell-free protein synthesis system (CFPS) was used to simultaneously overexpress CsAlaDC and PtGS to produce 3.82 mmol/L of theanine using alanine and glutamate as substrates, demonstrating that CsAlaDC can be used for theanine synthesis [63]. In the same year, Hagihara et al. [46] constructed two routes of de novo synthesis from glucose to theanine (Figure 2). One was the AlaDC pathway, which is a plasmid that simultaneously expresses *Pseudomonas syringae*-derived PsGMAS, CsAlaDC, and *B. subtilis* 168-derived pyruvate dehydrogenase BsAld in *E. coli*, and 1.53 g/L of theanine was produced [46]. Another pathway, known as the TA pathway, used plasmids to simultaneously express PsGMAS, BsAld, PpTA8, and the endogenous acetaldehyde dehydrogenase EutE in *E. coli.* However, the L-theanine yield (>300 mg/L) was lower than the AlaDC pathway [46]. Despite the low yield, this study boldly provided an effective method for producing L-theanine without supplemental ethylamine.

### 3.2. Glutamine-Mediated Pathway of L-Theanine in Microorganisms

In the glutamine-mediated pathway, L-glutaminase (GLS) and GGT play a major catalytic role, catalyzing the synthesis of L-theanine from Gln and ethylamine. Unlike the glutamate-mediated synthetic pathway, this pathway does not require ATP [7]. Tachiki et al. [47] first isolated GLS from *P. nitroreducens* IFO 12694 and found that it can

use hydroxylamine, methylamine, and ethylamine as receptor molecules, demonstrating that GLS from *Pseudomonas species* can synthesize theanine. New research has recently provided an efficient method for the production of theanine by the permeabilization of *P. nitroreducens* [48]. *P. nitroreducens* SP.001 cells were treated mainly with 15.5% sucrose solution and permeabilized to obtain a highly active GLS, which produced 85.358 g/L of theanine catalyzed by 1 U/mL of the enzyme, a conversion rate of 66.1% [48]. In addition, GLS from the source fungus *Trichoderma koningii* could also catalyze the synthesis of theanine. A total of 7.491 g/L of L-theanine was obtained with the addition of 3 mL of enzyme solution, which catalyzed 0.3 mol/L of L-glutamine and 0.9 mol/L of ethylamine; however, this conversion rate was low [49] (Table 1).

GGT and GLS are found in a range of eukaryotes and prokaryotes and can catalyze both γ-glutamyl peptide hydrolysis and transpeptide reactions; however, mostly only *E. coli* and Bacillus species-derived GGT can be used in the synthesis of theanine [33,49]. The initial demonstration of GGT's ability to produce L-theanine was the discovery by Suzuji et al. [50] of *E. coli* K-12-derived E*cGGT*, with a 60% conversion rate for the synthesis of theanine from Gln and ethylamine catalyzed by the pure enzyme. Jia et al. [51] constructed an *E. coli* engineered bacterium with 26-times more crude enzyme solution activity under induction than the starting strain; however, Gln apparently did not convert as well as the pure enzyme catalysis. In addition, when using recombinant GGT from *Bacillus licheniformis* ER-15 for L-theanine biosynthesis, controlled univariate conversions of Gln were approximately 85–87% within 4 h; the immobilization of the recombinant enzyme in calcium alginate could also achieve a similar conversion [52]. Xu et al. [53] used B-FITTER software to analyze all amino acid residues in *EcGGT* in relation to temperature factors, and screened for a mutant enzyme with significantly improved thermal stability (E387Q). The mutant enzyme catalyzed 120 mmol/L of Gln to produce 18.51 g/L of theanine under 100 W ultrasound conditions, which was 2.61-times the yield of theanine without ultrasound treatment [53]. The recombinant expression of *E. coli* GGT was enhanced using a small ubiquitin-related modifier (SUMO) fusion technique; the yield of L-theanine was increased to 41 g/L, with a conversion rate of approximately 80% [54]. *Bacillus subtilis* was significantly superior to *E. coli* in terms of the secretion of heterologous enzymes and proproteins. By overexpressing the PrsA lipoprotein and improving the mRNA stability of the *Bacillus subtilis* ggt gene, a yield of L-theanine of 53 g/L and a glutamyl conversion of 74% were achieved in the optimized system [74]. The expression of recombinant ggt in a subspecies of *Bacillus glutamicus*, followed by the use of a tac promoter with an optimized sequence in the plasmid, significantly increased the activity of the ggt gene, resulting in a high transformation rate of 89.76% [56]. In addition, it was shown that *Luteibacter*, an endophytic bacterium isolated from tea tree, can also convert Gln and other amino acids into theanine in the absence of ethylamine (Figure 2). Although the yield was only 31.875 μg/L, it also indirectly suggests that there may be another microbial synthesis pathways for theanine production that is not dependent on ethylamine [58].

### 3.3. Other Substrate-Mediated Pathways of L-Theanine in Microorganisms

Synthetic pathways mediated by other glutamyl compounds as donors have been shown to exist, and are also capable of producing L-theanine when catalyzed by immobilized cells with GGT activity [10] (Figure 2). These glutamyl compounds include γ-Glutamyl-*p*-nitroanilide [75,76], glutathione (GSH) [59], glutamic acid γ-methyl ester (GMAE) [10], L-glutamine-Zn(II) (Zn(Gln)$_2$) [77], and γ-L-Glutamylhydrazide [59]. Especially when GMAE is used as a novel substrate, the relative activity of GGT is 85.4%, which is 1.2-times higher than the activity of GGT with Gln as substrate [59]. Moreover, when the ratio of GMAE to ethylamine is 1:12, the conversion rate can reach as high as 96.3% [59].

### 4. Conclusions

At present, L-theanine is used in a wide range of applications in food, pharmaceuticals, health care, and cosmetics. The traditional chemical process for the synthesis of L-theanine

is environmentally unfriendly and energy consuming. With the development of synthetic biology, the industrial fermentation of L-theanine has become an inevitable trend in the pursuit of effective and safe production. This review summarized the synthesis pathways of L-theanine in different plants and microorganisms, and explored the different methods used in the microbial synthesis of L-theanine, which can be divided into three main groups. The first is the enzyme-catalyzed preparation method, which uses an extracted crude or pure enzyme solution to catalyze the reaction. The second is the whole-cell catalyzed preparation method, which is a process that uses intact microbial cells to convert the substrate into L-theanine. Additionally, the third is the microbial fermentation method, which is divided into two main methods: ethylamine flow addition and glucose de novo fermentation, in which different carbon sources are used in the ethylamine flow addition method as well as a strategy of carbon source superposition to broaden the selectivity of the substrate. It is clear that microbial fermentation is more economical and more in line with contemporary concepts of safe production. Two main categories are distinguished by different precursor-mediated approaches: divided into Glu-mediated L-theanine syn and Gln-mediated L-theanine synthesis pathways. With their different key catalytic enzymes and the fact that the Gln pathway does not require ATP, the other substrate in the Glu-mediated synthetic L-theanine pathway can be replaced by other amino acids in addition to ethylamine, such as lysine. Synthetic pathways mediated by other glutamyl compounds as donors are still in the early stages of research, but have potential research implications, such as GMAE.

Theanine in plants is mostly synthesized in tea tree; however, some plants, such as *Z. mays*, also contain genes and enzymes that can catalyze the production of theanine. However, due to the low yield of theanine in plants and the complexity of extraction and purification, it is difficult to meet the requirements for the large-scale production of theanine. In the industrial development of the microbial synthesis of L-theanine, the ultimate aim is to achieve the maximum production of the target product at the lowest economic cost and to maximize productivity. In order to achieve this, the following proposals were created on the basis of the research of many scholars. Firstly, the natural synthesis pathway for L-theanine synthesis in plants was investigated through bioinformatics to uncover and characterize key enzyme genes. Secondly, microbial fermentation was the most promising of the microbial synthesis methods, and the one-step fermentation of glucose is worthy of further study, such as optimizing the metabolic fluxes of the existing TA and AlaDC pathways and enhancing the function and annotation clarity of enzymes in the de novo synthesis pathway. Thirdly, in the Glu-mediated synthetic pathway, the production of L-theanine by GMAS was better facilitated by improving an efficient ATP regeneration system. Fourthly, the production of L-theanine was oriented towards food grade and industrialization; therefore, in-depth studies on the expression of food-grade enzymes and further studies on targeted mutagenesis and molecular modifications to improve the safety and activity of the enzymes should be performed.

**Author Contributions:** Writing—original draft preparation, S.X. and R.Q.; writing—review and editing, S.H.; visualization, Z.F.; supervision, T.B. and W.W.; project administration, J.Z. and J.C.; funding acquisition, J.Z. and J.C. All authors have read and agreed to the published version of the manuscript.

**Funding:** This work was supported by the National Natural Science Foundation of China (22108017) and the Natural Science Foundation of Sichuan Province (2022NSFSC1614).

**Institutional Review Board Statement:** Not applicable.

**Informed Consent Statement:** Not applicable.

**Data Availability Statement:** Not applicable.

**Conflicts of Interest:** The authors declare no conflict of interest.

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
