# Peer review of "Advances in the Production of Theanine by Plants and Microorganisms"

_fermentation, doi:10.3390/fermentation9060543_

Round 1
Reviewer 1 Report
Very good review! Informative and well structured.
A few typos and gramaticaly incorrect sentences but good english overall.
Author Response
Reviewer #1
Comments and Suggestions for Authors: Very good review! Informative and well structured.
Comments on the Quality of English Language: A few typos and gramaticaly incorrect sentences but good english overall.
Response: Thank you for your affirmation and comments. Now the language have been revised in the manuscript.
Reviewer 2 Report
Please see the attachment

Line 366: the sentence "And the third is the microbial fermentation ..." should be reorganized.
– Line 31: "... antioxidant ageing, anti-inflammatory and anti-tumour." This last part of the sentence does not fit its first part ("its synergistic ability to ..."). "Antioxidant ageing" could be understood as the aging of antioxidant (as a substance).
– Line 150: a title of Section 3 "The microbial synthesis pathway of L-theanine in microorganisms" should contain either "microbial" or "in microorganisms".
Author Response
Reviewer #2
General comments to the manuscript
- The manuscript fits ideally the aims and scope of the Special issue "New Insights intoAmino Acid Biosynthesis". However, a rather large amount of recent reviews are concerned withbiosynthesis of theanine, as exemplified below. That is why the authors should specify the originality of the present review. It can be made without hard efforts. A kind of "Novelty statement" should be presented within the framework of Introduction, which describes the originality of manuscript. The previous published reviews on the problem should be cited in this subsection and throughout the manuscript where appropriate. May be, new references have to be introduced in the discussion within the manuscript.
Response: Thank you for your affirmation and comments.
In lines 68-70 on page 2, the sentence “In addition, microbial production of L-theanine can replace traditional non environ-mentally friendly and energy intensive methods, and is worth trying to apply in vari-ous industries” was added. And the following four references have been added to the appropriate positions in the introduction section:
Chen, Z., Wang, Z., Yuan, H., & He, N. (2021). From tea leaves to factories: a review of research progress in L-Theanine biosynthesis and production. Journal of Agricultural and Food Chemistry, 69(4), 1187-1196.
Lin, S., Chen, Z., Chen, T., Deng, W., Wan, X., & Zhang, Z. (2023). Theanine metabolism and transport in tea plants (Camellia sinensis L.): advances and perspectives. Critical Reviews in Biotechnology, 43(3), 327-341.
Zhao, J., Li, P., Xia, T., & Wan, X. (2020). Exploring plant metabolic genomics: chemical diversity, metabolic complexity in the biosynthesis and transport of specialized metabolites with the tea plant as a model. Critica lRreviews in Biotechnology, 40(5), 667-688.
Liu, S. H., Li, J., Huang, J. A., Liu, Z. H., & Xiong, L. G. (2021). New advances in genetic engineering for L-theanine biosynthesis. Trends in Food Science & Technology, 114, 540-551.
- This review is intended principally to provide theoretical support for the industrializationof theanine (Line 71). Therefore, a presumable line of discussion within the manuscript favoring its originality could consist in chemical backgrounds of this amino acid. Theanine is γ-glutamyl-Lethylamide, and a risk of ethylamine occurrence as a product of theanine transformation exists. The authors should try to assess risks of the foods artificially enriched with theanine (even though thelatter is of the biotechnological, not chemical, origin).
Response: Thank you for your affirmation and comments. Now we have revised our manuscript according to your constructive comments.
- "Conclusions" section contains some inconsistencies. Line 366: the sentence "And thethird is the microbial fermentation ..." should be reorganized. Besides, "Conclusions" are devoted almost entirely to the industrial development of the microbial synthesis of L-theanine, whereas a title of the manuscript tells us about biosynthesis by plants and microorganisms.
Response: Thank you for your affirmation and comments. Now we have revised our manuscript according to your constructive comments.
In lines 372-373 on page 10, “And the third is the microbial fermentation method, which is divided into two main methods that additional flow addition of ethylamine and ab initio fermentation of glucose.” was revised as “And the third is the microbial fermentation method, which is divided into two main methods: ethylamine flow addition and glucose de novo fermentation.”
On pages 10 and 11, a description of the synthesis of theanine by plants has been added to the 'Conclusion'
Proposed corrections to manuscript
– Line 31: "... antioxidant ageing, anti-inflammatory and anti-tumour." This last part of the sentence does not fit its first part ("its synergistic ability to ..."). "Antioxidant ageing" could be understood as the aging of antioxidant (as a substance).
– Line 150: a title of Section 3 "The microbial synthesis pathway of L-theanine in microorganisms" should contain either "microbial" or "in microorganisms".
Response: Thank you for your affirmation and comments. Now we have revised our manuscript according to your constructive comments.
In line 31 on page, “antioxidant ageing” was revised as “anti-ageing”.
In line 150 on page 5, “The microbial synthesis pathway of L-theanine in microorganisms” was revised as “The synthesis pathway of L-theanine in microorganisms”
Chen, Z., Wang, Z., Yuan, H., & He, N. (2021). From tea leaves to factories: a review of research progress in L-Theanine biosynthesis and production. Journal of Agricultural and Food Chemistry, 69(4), 1187-1196.
Lin, S., Chen, Z., Chen, T., Deng, W., Wan, X., & Zhang, Z. (2023). Theanine metabolism and transport in tea plants (Camellia sinensis L.): advances and perspectives. Critical Reviews in Biotechnology, 43(3), 327-341.
Zhao, J., Li, P., Xia, T., & Wan, X. (2020). Exploring plant metabolic genomics: chemical diversity, metabolic complexity in the biosynthesis and transport of specialized metabolites with the tea plant as a model. Critica lRreviews in Biotechnology, 40(5), 667-688.
Liu, S. H., Li, J., Huang, J. A., Liu, Z. H., & Xiong, L. G. (2021). New advances in genetic engineering for L-theanine biosynthesis. Trends in Food Science & Technology, 114, 540-551.